# Preventive Effect of Cocoa Flavonoids via Suppression of Oxidative Stress-Induced Apoptosis in Auditory Senescent Cells

**DOI:** 10.3390/antiox11081450

**Published:** 2022-07-26

**Authors:** Luz del Mar Rivas-Chacón, Joaquín Yanes-Díaz, Beatriz de Lucas, Juan Ignacio Riestra-Ayora, Raquel Madrid-García, Ricardo Sanz-Fernández, Carolina Sánchez-Rodríguez

**Affiliations:** 1Department Clinical Analysis, Hospital Universitario de Getafe, Carretera de Toledo, Km 12.5, 28905 Getafe, Madrid, Spain; luzmar.rivas@salud.madrid.org; 2Department Otolaryngology, Hospital Universitario de Getafe, Carretera de Toledo, Km 12.5, 28905 Getafe, Madrid, Spain; joaquin.yanes@salud.madrid.org (J.Y.-D.); juanignacio.riestra@salud.madrid.org (J.I.R.-A.); rsanzf@salud.madrid.org (R.S.-F.); 3Department of Medicine, Faculty of Biomedical and Health Sciences, Universidad Europea de Madrid, 28670 Madrid, Spain; beatriz.delucas@universidadeuropea.es (B.d.L.); raquel.madrid@universidadeuropea.es (R.M.-G.)

**Keywords:** age-related hearing loss, cocoa, antioxidant agents, apoptosis, mitochondrial-apoptotic pathway, viability, senescence, oxidative stress

## Abstract

Presbycusis or Age-related hearing loss (ARHL) is a sensorineural hearing loss that affects communication, leading to depression and social isolation. Currently, there are no effective treatments against ARHL. It is known that cocoa products have high levels of polyphenol content (mainly flavonoids), that are potent anti-inflammatory and antioxidant agents with proven benefits for health. The objective is to determine the protective effect of cocoa at the cellular and molecular levels in Presbycusis. For in vitro study, we used House Ear Institute-Organ of Corti 1 (HEI-OC1), stria vascularis (SV-k1), and organ of Corti (OC-k3) cells (derived from the auditory organ of a transgenic mouse). Each cell line was divided into a control group (CTR) and an H_2_O_2_ group (induction of senescence by an oxygen radical). Additionally, every group of every cell line was treated with the cocoa polyphenolic extract (CPE), measuring different markers of apoptosis, viability, the activity of antioxidant enzymes, and oxidative/nitrosative stress. The data show an increase of reactive oxidative and nitrogen species (ROS and RNS, respectively) in senescent cells compared to control ones. CPE treatment effectively reduced these high levels and correlated with a significant reduction in apoptosis cells by inhibiting the mitochondrial-apoptotic pathway. Furthermore, in senescence cells, the activity of antioxidant enzymes (Superoxide dismutase, SOD; Catalase, CAT; and Glutathione peroxidase, GPx) was recovered after CPE treatment. Administration of CPE also decreased oxidative DNA damage in the auditory senescent cells. In conclusion, CPE inhibits the activation of senescence-related apoptotic signaling by decreasing oxidative stress in auditory senescent cells.

## 1. Introduction

Aging is the natural physiological process that is characterized by progressive degenerative changes in most organs and tissues of organisms, including damage to the hearing organ [1]. This process is associated with degeneration of the auditory function, age-related hearing loss (ARHL) or presbycusis [2]. ARHL is one of the most frequent diseases affecting the elderly population; according to the World Health Organization, around one-third of people over 65 years of age are affected by ARHL [3]. This condition is characterized by a decrease in hearing sensitivity and speech understanding in noisy environments, among others. ARHL is symmetrical bilateral hearing loss, progressive, irreversible, and produces degeneration of the cochlea, resulting from either loss of sensory hair cells or loss of auditory nerve fibers during cochlear aging [1,2,4].

To date, the mechanism underlying ARHL is not well understood. ARHL is a multifactorial disorder to which numerous risk factors contribute, both extrinsic (noise, trauma, exposures to environmental ototoxic agents, metabolic changes, vascular lesions, diet, and the immune system) and intrinsic (genetic factor and the physiological process of aging) [5,6]. The accumulation of the effects of these factors causes the development of ARHL.

Oxidative injury induced by free radicals is presumably the principal cause of age- associated pathology in the biological aging of cells. Therefore, oxidative damage is an essential intrinsic factor in the pathogenesis of ARHL. The increased levels of free radicals (reactive oxygen species [ROS] and reactive nitrogen species [RNS]) act as inductors of oxidative stress and damage [6]. Mitochondria generates ROS in mammalian cells and plays a crucial role in aging as a leading source of ROS [7,8]. ROS causes damage to mitochondrial components, such as respiratory chain proteins, mitochondrial membranes, mitochondrial DNA (mtDNA) and nuclear DNA that affect mitochondrial and cellular function, which lead to the triggering of apoptotic cell death pathways of auditory cells [7,8,9]. Oxidative stress is an imbalance between ROS production and endogenous antioxidant levels. Under normal conditions, ROS are scavenged or metabolized by endogenous antioxidant mechanisms (e.g., superoxide dismutase [SOD], glutathione peroxidase [GPx], and catalase [CAT]) and balance inner ear homeostasis [10,11]. However, the aging process alters this homeostasis condition in the hearing organ.

Once there is a loss of spiral ganglion neurons and hair cells in ARHL, the recovery of hearing loss is impossible because these cell types do not regenerate [2]; therefore, the prevention of ARHL is critical. In the last years, it has been found that targeting oxidative stress by pharmacological treatment can help to slow or prevent ARHL. Antioxidant food constituents have a protective role against oxidative stress-induced degenerative and age-related diseases [2,12]. Among these are plant polyphenols (flavonoids), which are present in fruits, vegetables, and beverages including wine and tea with important biological activities such as anti-inflammatory, antioxidant, anticarcinogenic, and antiviral ones [12,13,14]. Most of the biological mechanisms of polyphenols have been linked to their antioxidant capacity and free radical scavenging. However, they may also act by increasing endogenous antioxidant defense [12,13,14].

Cocoa beans from the *Theobroma cacao* (cocoa tree) are an essential source of polyphenols (flavonoids) and more specifically of flavanols. Cocoa-derived products are habitually consumed in the majority of countries in the World [15,16]. Thus, they can be considered as natural products that provide dietary antioxidants with therapeutic qualities. Many studies have shown that cocoa possesses beneficial effects against oxidative stress-related diseases by increasing the activities of antioxidant enzymes and by scavenging free radicals [15,16]. However, to the best of our knowledge, chemopreventive effects exerted by cocoa polyphenols on ARHL have not yet been investigated.

We have recently shown that polyphenol mixtures of tannic acid, resveratrol, quercetin, rutin, gallic acid, and morin have the ability to protect against ARHL in rats. These polyphenols generated significant protection against ARHL in Sprague–Dawley (SD) rats, with significantly improved ASSR (Auditory Steady-State Response) and tone-burst ABR (Auditory Brainstem Response) auditory thresholds in rats receiving treatment with polyphenols [17]. In addition, we demonstrated that treatment with polyphenol mixtures inhibits the activation of age-related apoptotic signaling by decreasing oxidative stress inside the rat cochlea [18,19].

The present work aimed to evaluate the underlying mechanisms involved in the protection exerted by cocoa polyphenolic extract (CPE) against damage induced by aging (induction of cellular senescence by H_2_O_2_). To this end, the impact of CPE treatment on ROS/RNS generation, apoptosis induction, mitochondrial-pathway and oxidative damage to DNA was investigated. The effect of cocoa polyphenols on the activities of GPx, CAT, and SOD-related enzymes and their role in CPE-induced cytoprotection were also evaluated. In summary, treatment with cocoa polyphenolic extract inhibits the activation of age-related apoptotic signaling by decreasing oxidative/nitrosative stress within auditory senescent cells.

## 2. Materials and Methods

### 2.1. Cell Culture

Auditory cell lines, OC-k3, SV-k1, and HEI-OC1, were provided by: Dr. Beatriz Duran Alonso (Institute of Molecular Biology and Genetics [IBGM], University of Valladolid, Valladolid, Spain), Dr. Federico Kalinec (House Research Institute, Los Angeles, CA, USA), and Dr. Maria Rosa Aguilar (Department of Nanomaterials and Polymeric Biomaterials Institute of Science and Technology of Polymers CSIC, Madrid, Spain), respectively. Cells were cultured in Dulbecco’s high glucose Eagle’s medium (DMEM; Gibco BRL, Waltham, MA, USA) supplemented only with 10% fetal bovine serum (FBS; Gibco BR, Waltham, MA, USA), at 10% CO_2_ and 33 °C, previously described [20].

### 2.2. Cocoa Polyphenol Extraction

High flavanol cocoa powder (Chococru, London, UK) was used for this study. To extract polyphenols from cocoa powder, we used a modified method, as previously described [21]. The polyphenols were extracted from 1 g of cocoa powder by washing with 40 mL of 0.8% hydrochloric acid 2 N in 50% aqueous methanol (50:50, *v*/*v*) for 1 h and 40 mL of acetone–water (70:30, *v*/*v*) for 1 h of shaking at room temperature. Then, the samples were centrifuged at 3000× *g* for 15 min and the supernatants of each extraction were combined. The desiccated extract by rotavapor R-14 (BüchiLabortechnik AG, Flawil, Switzerland) was dissolved in distilled water, and the extract was lyophilized in LyoQuestlyophilizer (Telstar, Terrassa, Spain). The BQC Phenolic Quantification Assay Kit (Bioquochem, Oviedo, Spain) based in Folin–Ciocalteau spectrophotometric method [22] was used to determine the total polyphenol content using gallic acid as standard and employing the FLUOstar Omega (BMG Labtech, Ortenberg, Germany) plate reader to measure it.

### 2.3. Experimental Design and Cell Treatments

Cellular senescence was induced by treating the cells with H_2_O_2_ (VWR Inc., West Chester, PA, USA) at concentrations of 100 μM for 1 h. Post-treatment analysis was performed at 24 h [23].

For the cocoa treatment, different concentrations of CPE (0.5, 5, 10 and 20 μg/mL) were diluted in a serum-free culture medium and filtered with a 0.2-μm membrane. Then, CPE was added to the cell cultures for 20 h. After the CPE treatment, a 100 μM of H_2_O_2_ concentration was applied to the auditory cells belonging to the H_2_O_2_ group for 1 h, to induce cellular senescence and evaluate the protective effect of the CPE against aging.

Figure 1 below describes the scheme of the experimental design used in this study.

### 2.4. Reactive Oxidative and Nitrogen Species Detection

The total free radicals were measured by the OxiSelect™ In Vitro ROS/RNS Assay Kit (Cell Biolabs Inc., San Diego, CA, USA). The assay employs a specific fluorescent probe, dichlorodihydrofluorescein DiOxyQ (DCFH-DiOxyQ). The DCFH-DiOxyQ probe can react with several ROS and RNS, such as peroxyl radical (ROO^•^), hydrogen peroxide (H_2_O_2_), peroxynitrite anion (ONOO^−^), and nitric oxide (NO). Fluorescence measurement was performed on a FLUOstar Omega (BMG Labtech, Ortenberg, Germany) plate reader (excitation to 485 nm and emission to 530 nm).

### 2.5. Reactive Oxygen Species Detection

Fluorescent probe dihydroethidium (DHE; Calbiochem, San Diego, CA, USA) was used to measure superoxide anion (O_2_^−^), peroxynitrite anion (ONOO^−^) or hydroxyl radical (^•^OH) generation [24,25]. Cells were seeded in the Lab-Tek II Chamber Slide System at 2 × 10^4^ cells/well. After 24 h, the cells were treated according to the experimental group and incubated for 24 h. Next, the cells were fixed with 4% paraformaldehyde (PFA) for 10 min and then were incubated with probe DHE (4 µmol/L) for 90 min at 37 °C. After this, the cell nuclei were stained with 300 nM of 4′,6-diamidino-2-phenylindole dihydrochloride (DAPI; Sigma-Aldrich, San Luis, CA, USA) for 5 min at 37 °C. The DAPI marks the cell nucleus by binding to the DNA. The samples were observed on an Olympus BX51 microscope. The images were analyzed by ImageJ software (Version 1.53j, Bethesda, MD, USA).

### 2.6. Determination of Oxidative DNA Damage

The Oxiselect Oxidative DNA Damage ELISA Kit (Cell Biolabs, Sna Diego, CA, USA) was used to quantify 8-hydroxy-20-deoxyguanosine (8-OHdG), a marker of oxidative DNA damage. The ELISA Kit is a competitive enzyme immunoassay. Genomic DNA from cells’ DNA samples was isolated using a genomic DNA extraction kit (Chemicon), following the manufacturer’s instructions. The absorbance was measured at 450 nm in a microplate reader FLUOstar Omega (BMG Labtech, Ortenberg, Germany). Concentrations of 8-OHdG were expressed as nanograms per milliliter.

### 2.7. Total Superoxide Dismutase (SOD) Activity Measurements

Total SOD activity (Cu/Zn-, Fe- and Mn-SOD) was determined in cell lysates by the SOD assay kit (Cayman Chemical Company, Ann Arbor, MI, USA) a colorimetric method, according to the manufacturer’s instructions. Color intensity was read using a FLUOstar Omega (BMG Labtech, Ortenberg, Germany) plate reader.

### 2.8. Glutathione Peroxidase (GPx) Activity Measurements

GPx activity was measured in cell lysates by the GPx Assay kit (Cayman Chemical Company, Ann Arbor, MI, USA), following the manufacturer’s instructions. Color intensity was determined by a FLUOstar Omega (BMG Labtech, Ortenberg, Germany) plate reader set to 340 nm.

### 2.9. Catalase (CAT) Activity Measurements

The CAT Assay kit (Cayman Chemical Company, Ann Arbor, MI, USA) determines enzyme activity by the peroxidic function of CAT. Color intensity was read in a FLUOstar Omega (BMG Labtech, Ortenberg, Germany) plate reader set to 540 nm.

### 2.10. Measurement of Total Antioxidant Capacity

The total antioxidant capacity (TAC) was measured by the e-BQC portable device (Bioquochem; Oviedo, Spain) in the cell culture supernatants. The system is based on the measurement of redox potential (charge/period or micro-Coulomb, μC). The samples (30 µL) were dispensed onto a disposable strip. System readings were given for rapid (Q1), slow (Q2), and total (QT: Q1 + Q2) antioxidant responses.

### 2.11. Indirect Immunofluorescence

After treatments, the cells were fixed (10 min in PFA 4%) and blocked (at 37 °C for 1 h). Then, to cells were added the antibody Annexin-V (1/200) (Abcam, Cambridge, UK) at 4 °C overnight. Subsequently, the samples were incubated with the secondary antibody Alexa Fluor 546 (1/250; Molecular Probes, Eugene, OR, USA) for 45 min at 37 °C. Finally, the cell nuclei were stained with DAPI (Sigma-Aldrich, San Luis, CA, USA) for 5 min at 37 °C. Cells were observed on an Olympus BX51 microscope, and the ImageJ software (Version 1.53j) was used to analyze the images. The suppression of the primary antibody evaluated the specificity.

### 2.12. Quantification of Caspases

Apoptosis was measured by CaspasesMultiplex Activity Assay Kit (Abcam, Cambridge, UK) in cell culture extracts according to the manufacturer’s protocol. Fluorescence was determined by the FLUOstar Omega (BMG Labtech, Ortenberg, Germany) microplate reader at: Ex/Em = 535/620 nm (Caspase 3); Ex/Em = 490/525 nm (Caspase 8); and Ex/Em = 370/450 nm (Caspase 9).

### 2.13. Measurement of ATP-Levels

The CellTiter-Glo^®^ Luminescent Cell Viability Assay is a method for the quantification of the Adenosine triphosphate (ATP), as an indicator of metabolically active cells. The ATP was determined by previously described methods [26]. Luminescence was read by a microplate reader FLUOstar Omega (BMG Labtech, Ortenberg, Germany). The levels of ATP were normalized to the protein content (10 µg total protein extract cell) and calculated based on the standard curve.

### 2.14. Quantitative Real-Time Reverse Transcription-Polymerase Chain Reaction

The miRNeasy Tissue/Cells Advanced Mini Kit (QIAGEN) was used to extract total RNA from the auditory cells by following the manufacturer’s protocol. The total RNA (1 μg) was reverse transcribed by a StaRT kit (AnyGenes, Paris, France) following the manufacturer’s protocol in a Veriti Thermal Cycler (Applied Biosystems, Waltham, MA, USA).

The 7500 Fast real-time PCR detection system (Applied Biosystems, Waltham, MA, USA) was performed to study gene expression. Complementary DNA (cDNA) (2 μL) was mixed with 8 μL of Perfect MasterMix SYBRG (AnyGenes, Paris, France). Polymerase chain reactions were performed according to the protocols of the manufacturer in triplicate. The reference genes used to normalize the transcript levels were β-actin and glyceraldehyde-3-phosphate dehydrogenase (GAPDH). The comparative “Ct” method was performed for the determination of the relative expression levels.

The Caspase-3, -8 and -9 (AnyGenes, Paris, France) human primers were used in this study.

### 2.15. Cytochrome c Quantification

The Rat/Mouse Cytochrome c Quantikine assay kit (R&D System, Minneapolis, MN, USA) was used for the quantitative determination of mouse cytochrome c concentrations in cell lysates. Color intensity was read by a FLUOstar Omega (BMG Labtech, Ortenberg, Germany) plate reader set to 450 nm.

### 2.16. Bax and Bcl-xL Quantification

The Mouse Bax ELISA Kit and Mouse/Rat Bcl-xLSimpleStep ELISA Kit (Abcam, Cambridge, UK) measure Bax and Bcl-xL levels, respectively, in cell lysates, following the manufacturer’s protocol. Color intensity was read by a FLUOstar Omega (BMG Labtech, Ortenberg, Germany) plate reader set to 450 nm.

### 2.17. Statistical Analysis

The statistical study was carried out by variance ANOVA test and Tukey’s multiple comparison test using the SPSS 19.0 software (IBM, Armonk, NY, USA). Data are represented as mean ± standard deviation (SD). *p* < 0.05 values indicate statistical significance.

## 3. Results

### 3.1. Total Polyphenol Content in Cocoa Extract

The total content of cocoa polyphenols obtained after the extraction process was 584.45 μg/mL. In addition, we analyzed the total content of polyphenols post-treatment in cell culture supernatants (Table 1). The results showed significant differences in total polyphenol content between cells treated and not treated with CEP. We observed that cells that were not treated with CEP, whether or not they were treated with H_2_O_2_, had a polyphenol concentration of 0 μg/mL, while in the groups treated with CPE, increasing concentrations of polyphenols were obtained as a function of the initial dose of CPE (Table 1).

### 3.2. Cocoa Inhibited H_2_O_2_-Induced Damage on Viability in Auditory Cells

As shown in Figure 2, H_2_O_2_ decreased cell viability in HEI-OC1, OC-k3, and SV-k1 cells. The cell viability was reduced to 57 ± 2.16% in HEI-OC1 cells, to 56 ± 1.56% in OC-K3 cells and to 67 ± 2% in SV-k1 cells, when treated with H_2_O_2_ (100 μM) for 1 h compared to the control group. However, CPE pretreatment significantly attenuated the detrimental effect of H_2_O_2_ on cell viability in auditory cells (*p* < 0.05; Figure 2). Also, CPE from 0.5 to 5 μg/mL alone had no notable effects on cell viability in auditory cells (Figure 2). At each experimental condition, 10 and 20 μg/mL of CPE produced a cytotoxic effect and, therefore, these high concentrations were not used for the subsequent tests.

### 3.3. Cocoa Decreased the Levels of Reactive Oxygen and Nitrogen Species in Auditory Senescence Cells

The next step was to determine the effect of CPE treatment on ROS and RNS levels in the cell extract of the auditory cells. Figure 3 shows that the levels of ROS/RNS in the CTR group (not H_2_O_2_-treated) are low, while the levels significantly increased in the H_2_O_2_ group (senescence). CPE treatment reduced the elevated levels of ROS/RNS in the H_2_O_2_ group of the three auditory cell lines (Figure 3).

The fluorescence DHE probe detects ROS by visualizing the increased intensity of red fluorescence in the nuclei of cells. The senescence cells in the H_2_O_2_ group showed an augmented fluorescence compared to the non-H_2_O_2_-treated group (CTR group) (Figure 4). After treatment with CPE, fluorescence intensity decreased in the H_2_O_2_ cells group, reflecting a reduction in the production of ROS but not in the treated CTR group (Figure 4).

### 3.4. Cocoa Extract Prevented Oxidative DNA Damage in Senescent Auditory Cells

Oxidative stress, which causes DNA damage, was measured through 8-hydroxy-20-deoxyguanosine (8-OHdG) formation in cells post-treatments (Figure 5). H_2_O_2_ treatment increased levels of 8-OHdG in three auditory cell lines (Figure 5). The data showed a significant difference in the levels of 8-OHdG associated with the treatment of CPE in the H_2_O_2_ group. This change was due to a significant decrease in 8-OHdG levels with CPE treatment of the H_2_O_2_ group (e.g., 5 μg of CPE showed: 3.4 ± 0.21 ng/mL, in HEI-OC1 cells; 3.0 ± 0.18 ng/mL, in OC-K3 cells; and 2.1 ± 0.ng/mL, in SV-k1 cells) when compared with the respective H_2_O_2_ groups without CPE treatment (e.g., HEI-OC1 cells showed 5.5 ± 0.2 ng/mL; OC-k3 cells showed 5.9 ± 0.18 ng/mL; and SV-k1 showed 4.4 ± 0.2 ng/mL) (Figure 5).

### 3.5. Cocoa Treatment Modulated Antioxidant Enzymes in Aged Auditory Cells

To determine the effect of CPE on three antioxidant systems in HEI-OC1, OC-k3, and SV-k1 cells, we measured SOD, CAT, and GPx activities with or without the CPE and H_2_O_2_ treatments. In H_2_O_2_-treated cells, there was a significant decrease in the total SOD, CAT, and GPx activities, compared with the CTR group (Figure 6). The treatment with CPE increased SOD activity (*p* < 0.05) in senescent cells (H_2_O_2_-treated cells) but not in young cells. In the case of CAT and GPx, CPE treatment also significantly elevated their activities in the aged cells group (Figure 6).

### 3.6. Cocoa Treatment Increased Total Antioxidant Capacity in Auditory Senescent Cells

The results of total antioxidant capacity (TAC) shown as Q1 (fast antioxidants), Q2 (slow antioxidants), and Qt (TAC) are presented in Figure 7. Statistical analysis provided significant differences between CTR and H_2_O_2_ groups; and in the H_2_O_2_ group, between treated and untreated with CPE for Q1, Q2, and Qt (*p*  <  0.5). As can be noted in Figure 7, Q1, Q2, and Qt were higher in the CTR group as compared to the H_2_O_2_ group without CPE, and therefore, the senescent cells exhibited lower TAC than the CTR group. CPE treatment induced statistically significant recovery of antioxidant capacity in senescent auditory cells.

### 3.7. Cocoa Prevented Apoptosis Activation in Aged Auditory Cells

The aged auditory cells showed an increase in fluorescence (Figure 8), which is related to an increase in Annexin-V (apoptosis marker), in the cells in the CTR group (Figure 8). After administration of CPE, fluorescence intensity decreased in the senescent cells group, thereby indicating a reduction in apoptosis that did not occur in the CTR cells (Figure 8).

In addition, results presented in Figure 9 show that caspase-3 activity increased in senescent auditory cells. Figure 9A shows that CPE treatment caused a significant decrease in H_2_O_2_-associated caspase-3 activation. Given these results, we conclude this relates to changes at the caspase-3 mRNA expression level. In H_2_O_2_-treated cells, there was a significant increase in the expression of caspase-3 mRNA relative to that observed in cells untreated by H_2_O_2_ (Figure 9B), whose expression was very low. Treatment with CPE decreased relative expression of caspase-3 mRNA (*p* < 0.05) in senescent auditory cells, but not in CTR cells (Figure 9B).

### 3.8. Cocoa Increased ATP Levels in Aged Auditory Cells

We also checked ATP production (as a measure of metabolic cell status). ATP levels were significantly reduced after the treatment of H_2_O_2_ and as expected, levels of ATP in senescent cells that received CPE treatment were significantly higher than in their respective controls (Figure 10). ATP levels did not change in the non-H_2_O_2_-treated CTR group cells, treated or not with CPE (Figure 10).

### 3.9. Cocoa Impaired Activation of the Mitochondrial Cell Death Pathway in Aged Auditory Cells

We further explored the signaling cell death pathways activated upstream of caspase-3 activation in auditory cells. We focused on two: caspase-8 (extrinsic cell death pathway) and caspase-9 (mitochondrial or intrinsic cell death pathway) that, when activated, culminates in Caspase-3 activation, resulting in apoptosis. Figure 11A shows that caspase-9 activity was clearly increased by H_2_O_2_ administration in cells. Likewise, CPE treatment impaired senescent-related caspase-9 activation. We also measured caspase-8 activity after CPE and H_2_O_2_ treatments and a non-significant effect was obtained for all experimental groups (senescent or CTR cells) (Figure 11C). Also, we determined caspase-9 and 8 mRNA expression levels, and similar results were obtained as in the case of enzymatic activities. Preventive CPE treatment decreased caspase-9 gene expression in senescent auditory cells (Figure 11B); while no significant differences were obtained between all the experimental groups for the expression of Caspase-8 (Figure 11D).

To confirm the participation of the intrinsic pathway in apoptotic signaling, we also determined Bcl-xL (an anti-apoptotic protein) and Bax (a pro-apoptotic protein) protein expressions. Figure 12A shows that Bcl-xL protein levels are higher in CTR auditory cells and CPE-treated senescent auditory cells. In contrast, Bax protein was increased by H_2_O_2_-treatment in auditory cells (Figure 12B). CPE treatment in aged cells significantly decreased the expression of Bax (Figure 12B). In addition, the release of cytochrome c was markedly stimulated in the presence of H_2_O_2_ in auditory cells in comparison with cells from the CTR group (Figure 12C). Consistent with these results, the cytochrome c was reduced in the presence of CPE in senescent cells; no differences were observed in CTR cells (Figure 12C).

## 4. Discussion

Cellular senescence has been hypothesized as promoting age-associated organism dysfunction as well as age-related diseases such as ARHL via oxidative stress. Natural antioxidants inhibiting ROS production are fundamental in protecting the ear from age-induced damage. In this sense, cocoa and its derivates have been shown to contain important antioxidants such as polyphenolic compounds [27]. This biological property points to cocoa polyphenols as interesting candidates for cellular protection. In this study, we show that CPE protects auditory senescent cells against apoptosis by diminution of ROS generation, modulating oxidative DNA damage, and the activities of antioxidant enzymes, such as SOD, GPx, and CAT. In addition, we established that the capacity of CPE to inhibit apoptosis of cells through inhibition of the mitochondrial-apoptotic pathway is directly involved in its protective mechanism. This research has demonstrated for the first time that CPE might serve as a potential otoprotective agent against ARHL.

Several previous studies demonstrated that H_2_O_2_-induced senescence in different cell lines is a valuable model for evaluating mechanisms of senescence, and offers potential anti-senescence therapeutic targets, such as the cytoprotective effect of natural antioxidants [28,29,30,31]. We previously developed an auditory cell model of senescence by using several concentrations of H_2_O_2_, 100 μM among others, as an inductor of premature senescence in auditory cells (HEI-OC1, OC-k3, and SV-k1) [23]. We demonstrated that: auditory cell viability is impaired; the population doubling rate decreases; β-galactosidase activity increases (a marker of senescence); there is ROS overproduction; and oxidative stress post-H_2_O_2_ treatment increases DNA damage.

Numerous studies have described how increased production of intracellular ROS levels is an important mechanism underlying aging-induced Hair cells (HCs) death in the cochleae, resulting in age-associated hearing impairment [32,33]. Excessive ROS reduces antioxidant defense, triggering the release of cytochrome c from mitochondria and caspase-3 pathway activation, and leading to apoptosis [34]. To determine whether CPE exerted its protective effect against senescence through reduced cellular oxidative stress, we examined ROS and RSN production by performing several cellular assays. Our data indicated that CPE significantly decreased the generation of cellular ROS/RSN induced by H_2_O_2_ (senescence inducer) in HEI-OC1, OC-k3, and SV-k1 cells, which in turn significantly inhibited apoptosis. Our results were in agreement with previous studies that demonstrated that CPE is a scavenger of ROS/RSN. For example, Corcuera et al., 2012, demonstrated that treating human HepG2 cells with polyphenol-rich cocoa significantly decreases the free radical production induced by mycotoxins [35]. In another study, acrylamide cytotoxicity was also neutralized by CPE (10 μg/mL) in intestinal Caco-2 cells by reducing ROS, increasing the activity of antioxidant enzymes, and inhibiting the apoptotic pathways [36]. Previously, studies reported the antioxidant effects of CPE (0.1–30 μg/mL) on SH-SY5Y cell line (human neuroblastoma) treated with H_2_O_2_ FeSO4 by down-regulation of ROS production [37]. The cocoa procyanidin fraction (1 and 5 µg/mL) also reported protective effects against ROS accumulation induced by the aldehyde 4-hydroxynonenal in rat pheochromocytoma PC12 cells [38,39].

The antioxidant defense system plays a pivotal role in the protection of the organism against oxidative stress and comprises enzymatic and non-enzymatic constituents. Changes in the activity of antioxidant enzymes are considered markers of the antioxidant response [40,41]. The common characteristic of the implicated enzymes is their function of scavenging ROS to maintain the balanced redox state of cells [40,41]. Therefore, we examined the effect of CPE on activities of antioxidant enzymes such as CAT, GPx, and SOD in auditory senescent cells. Superoxide anions were generated during cellular respiration and converted to hydrogen peroxide (H_2_O_2_) by a reaction catalyzed by the enzyme SOD [40]. The CAT enzyme converts H_2_O_2_ into molecular oxygen and water, and GPx can use GSH as a reductant to catalyze H_2_O_2_ or organic hydroperoxides into water or the corresponding alcohols, respectively [40]. In our study, pretreatment with CPE in the senescent cells significantly increased SOD, CAT, and GPx activities. These data suggested that CPE activates cell survival signaling pathways, leading to an enhancement of the antioxidant defense system to eliminate free radicals. These results are similar to a previous report (Martin et al., 2007), where 2 or 20 h pretreatment of HepG2 cells with different concentrations of CPE (0.05–50 μg/mL) significantly inhibits cellular damage caused by tert-butyl hydroperoxide through the modulation of the activities of antioxidant enzymes [42]. Comparable results were obtained using the same oxidative stress model in pancreatic Ins-1E cells treated with 5–20 μg/mL of CPE for 20 h [43]. In another study performed on HepG2 cells, the beneficial effects of CPE against high-glucose-induced oxidative stress by modulating the antioxidant enzymes were reported [44]. Also, Yu et al. 2009 [45] showed that ursolic acid (extract of *Cornus officinalis* fruits) protects from hydrogen peroxide-induced HEI-OC1 damage through the activation of the antioxidant enzymes CAT and GPx. In vivo studies have also reported a connection between the consumption of cocoa and a reduction in different oxidative stress markers, such as antioxidant enzymes. Cocoa-enriched diets modify the SOD and CAT activities in the macrophages of heat-killed *Mycobacterium butyricum* suspension-induced arthritis in rat [46]. In another study, Nwichi et al. [47] reported improvement in the antioxidant defenses following 8 weeks of treatment with CPE in an animal model of spontaneous familial hypercholesterolemia.

We next evaluated the non-enzymatic antioxidant network by determination of the Total Antioxidant Capacity (TAC) of CPE treatment. We established that CPE increased TAC in the cultures of senescent cells. Data indicated a correlation between antioxidant defense and oxidative biomarkers. Chocolate presents very high levels of in vitro TAC compared to other foods of vegetable origin [48]. It has been shown to increase plasma TAC after ingestion in the short-term in acute intervention studies in hypercholesterolaemic individuals [49,50]. Human studies have determined that ingesting of plant foods such as wine, lettuce, tea, blueberries, and chocolate can modulate in vivo TAC [51].

Oxidative stress has been widely reported as inducing apoptosis of various types of cell populations in the cochlea associated with aging, eventually causing hearing loss [8,52,53]. The mitochondria-apoptotic pathway is regulated by the conjunct actions of the anti- and pro-apoptotic members of the Bcl-2 family of proteins, including activation of Caspase-9, and cytochrome c release, among others. Caspase-3 is the final effector in the mitochondria-mediated apoptosis [54]. Our results showed that exposure to H_2_O_2_ induced a significant increase of Caspase-3 and -9, up-regulation of the pro-apoptotic proteins Bax, down-regulation of the anti-apoptotic protein Bcl-xL, and a decrease in cytochrome c levels, whereas CPE significantly reversed these effects. These findings agreed with the reported beneficial effect of CPE on HepG2 cell apoptosis, whereby pretreatment with CPE before the oxidative injure blocked caspase-3 activation induced by t-BOOH [21]. Additionally, Kim et al. [55] showed that epigallocatechin-3-gallate (EGCG), the main component of green tea polyphenols, protects against NO-induced ototoxicity by suppressing the activation of caspase-3 in HEI-OC1 cells. We have recently demonstrated the benefits of a mixture of polyphenols in the ARHL rat model, which protected the cochlea from oxidant-induced apoptosis associated with age. This anti-apoptotic protection was mediated by a mechanism related to Bcl-2 activation (anti-apoptotic) and blocking of Bax (pro-apoptotic) and p53 [19].

In general, it is accepted that ROS may induce irreversible DNA damage [56]. Above 100 oxidative modifications to DNA have been identified, including both single- and double-strand breaks and adducts [57]. This DNA damage can be prevented by antioxidant treatments, as shown for example by Benkafadar et al. [58] who demonstrated that EUK-207 (synthetic superoxide dismutase/catalase mimetic) attenuated H_2_O_2_-induced DNA damage and senescence phenotype in HEI-OC1 cells and reduced loss of both hearing and hair cells in SAMP8 mice. In addition, Gutierrez-Mariscal et al. [59] support that consumption of a Mediterranean diet plus Coenzyme Q10 (a powerful antioxidant) reduces oxidative DNA damage in elderly subjects and decreases processes of cellular oxidation. Previous data from our group in an aged rat model show that the mixed polyphenol diet decreases the levels of oxidative DNA damage in the cochlea of aged rats (24 months of age) when compared to aged control rats. Corroborating this, we showed that decreased oxidative DNA damage in the auditory senescent cells with preventive CPE treatment contributes to inhibiting apoptosis.

## 5. Conclusions

In conclusion, we demonstrated that CPE exhibited a significant effect on protection against H_2_O_2_ (senescence inducer) insult by suppressing the generation of ROS/RNS and the mitochondrial apoptotic pathway in auditory cells. Our findings demonstrate for the first time the effectiveness of preventive CPE treatment against senescence-associated cell damage in three types of auditory cell populations. In summary, our data support previous results on the antioxidant effect of cocoa polyphenols. Therefore, cocoa products may contribute to the protection against diseases associated with oxidative stress (contributory or causal factor), such as ARHL. However, due to the limitations of in vitro models, in the future, additional preclinical (in vivo model) and clinical studies will be necessary to provide a clinical approach for patients.

## Figures and Tables

**Figure 1 antioxidants-11-01450-f001:**
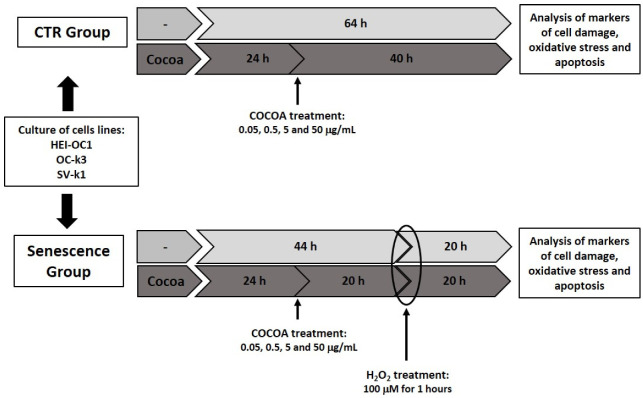
Comprehensive overview of the experimental timeline and procedures.

**Figure 2 antioxidants-11-01450-f002:**
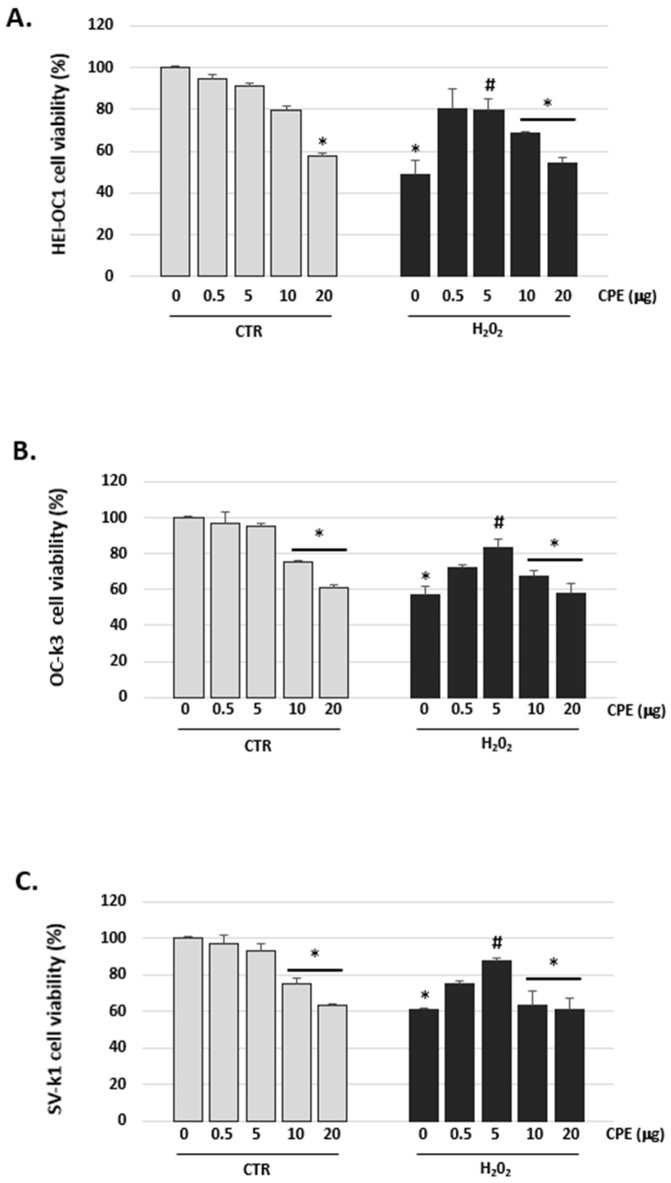
**Cocoa increased the viability of senescence auditory cells.** Cell viability measured by PrestoBlue Assay of auditory cells post treatments in (**A**) HEI-OC1, (**B**) OC-k3 and (**C**) SV-k1 cells. Experimental groups: CTR group (0.5; 5; 10 and 20 μg/mL of CPE) and H_2_O_2_ group (induced cellular senescence with 100 μM for 1 h in all groups: 0.5; 5; 10 and 20 μg/mL of CPE). All values are means ± SD (*n* = 4), and the analysis of variance (ANOVA) results. * *p* < 0.05 vs. CTR group; # *p* < 0.05 vs. H_2_O_2_ group.

**Figure 3 antioxidants-11-01450-f003:**
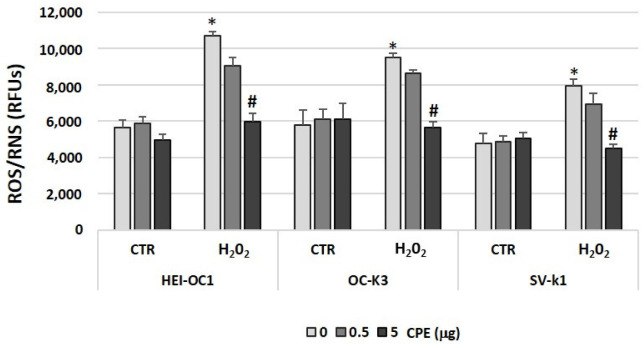
**Cocoa modulated total ROS/RNS productions in senescence auditory cells.** ROS/RNS were measured in HEI-OC1, OC-k3, and SV-k1 cells. Experimental groups: CTR group (0, 0.5 and 5 μg/mL of CPE for 20 h) and H_2_O_2_ group (induced cellular senescence with 100 μM for 1 h in all groups: 0, 0.5 and 5 μg/mL of CPE for 20 h). The diagrams represent the mean ± SD of relative fluorescent units (RFU). * *p* < 0.05 vs. CTR group; # *p* < 0.05 vs. H_2_O_2_ group (*n* = 4).

**Figure 4 antioxidants-11-01450-f004:**
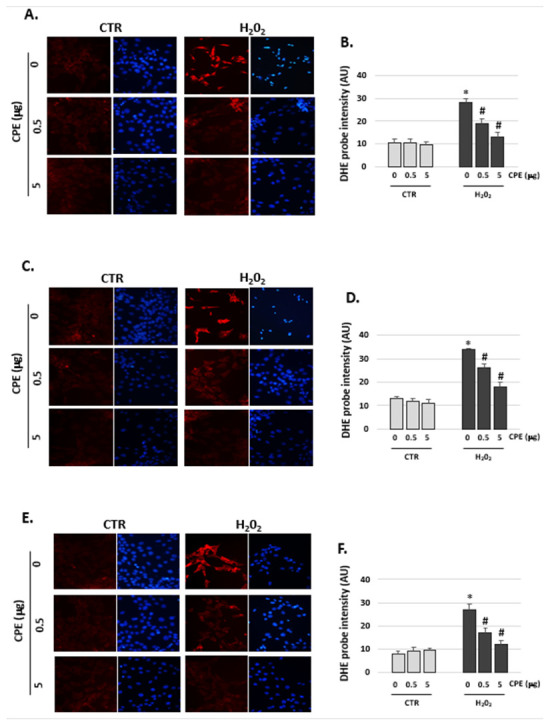
**Cocoa decreased the production of ROS in senescence auditory cells.** Representative images of fluorescence micrographs of (**A**) HEI-OC1, (**C**) OC-k3 and (**E**) SV-k1 cells (×40). ROS detection with the fluorescent probe dihydroethidium, DHE (red fluorescent), and DAPI (a marker of cell nuclei with blue fluorescent) dye in cell cultures at the end of the experiments. Experimental groups: CTR group (0; 0.5 and 5 μg/mL of CPE) and H_2_O_2_ group (induced cellular senescence with 100 µM for 1 h in all groups: 0; 0.5 and 5 μg/mL of CPE).Quantification of fluorescence intensity for DHE probe by ImageJ with (**B**) HEI-OC1, (**D**) OC-k3 and (**F**) SV-k1 cells. The diagrams represent the mean ± SD of Arbitrary units (AU). * *p* < 0.05 vs. CTR group; # *p* < 0.05 vs. H_2_O_2_ group (*n* = 4).

**Figure 5 antioxidants-11-01450-f005:**
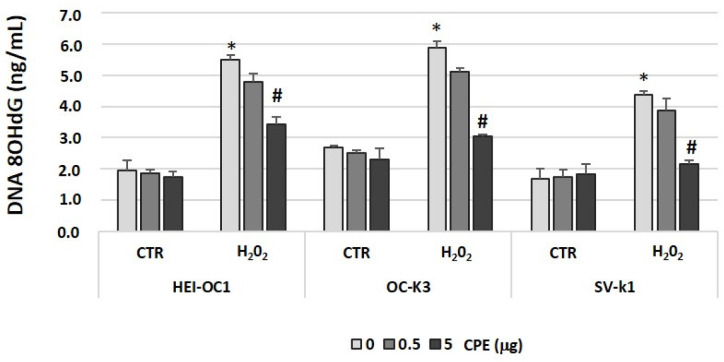
**Cocoa extract prevented H_2_O_2_-induced DNA oxidative damage in senescence auditory cells.** The 8-OHdG activities were measured by a colorimetric assay in protein lysates of HEI-OC1, OC-k3, and SV-k1 cells post-treatments. Experimental groups: CTR group (0, 0.5 and 5 μg/mL of CPE for 20 h) and H_2_O_2_ group (induced cellular senescence with 100µM for 1 h in all groups: 0, 0.5 and 5 μg/mL of CPE for 20 h). The values represent the mean ± SD in triplicate. * *p* < 0.05 vs. CTR group; # *p* < 0.05 vs. H_2_O_2_ group (*n* = 4).

**Figure 6 antioxidants-11-01450-f006:**
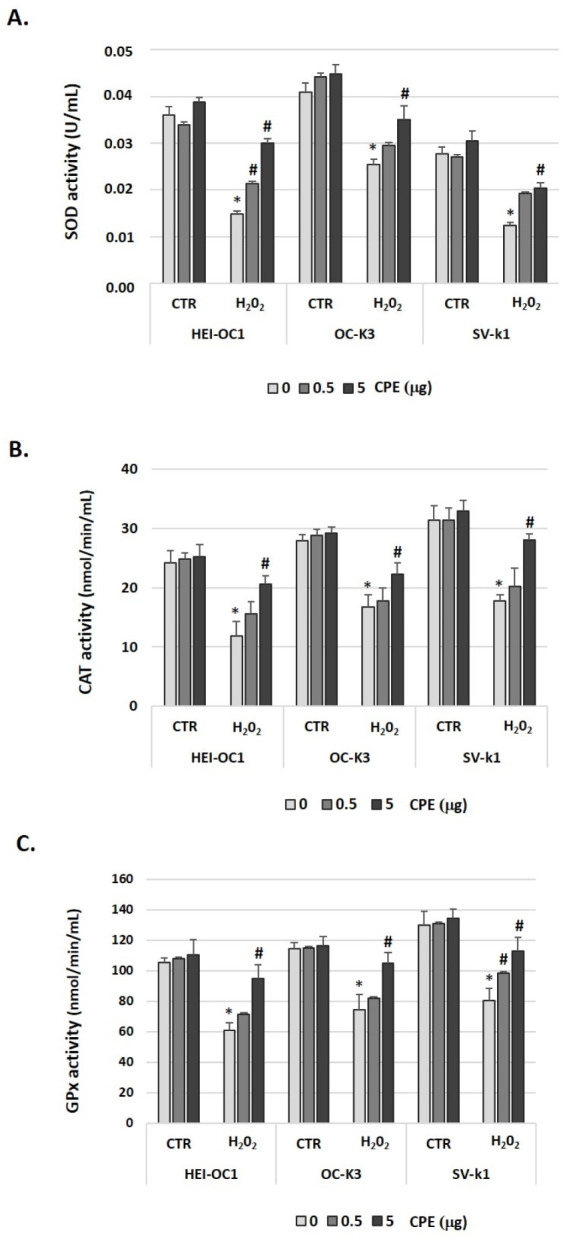
**Cocoa treatment modulated antioxidant enzymes in senescence auditory cells.** (**A**) SOD, (**B**) CAT, and (**C**) GPx activities were measured by a colorimetric assay in cell lysates of HEI-OC1, OC-k3, and SV-k1 cells post-treatments. Experimental groups: CTR group (0, 0.5 and 5 μg/mL of CPE for 20 h) and H_2_O_2_ group (induced cellular senescence with 100 µM for 1 h in all groups: 0, 0.5 and 5 μg/mL of CPE for 20 h). The values represent the mean ± SD of Units/mL (U/mL) (**A**) and (nmol/min/mL) (**B**,**C**) in triplicate. * *p* < 0.05 vs. CTR group; # *p* < 0.05 vs. H_2_O_2_ group (*n* = 4).

**Figure 7 antioxidants-11-01450-f007:**
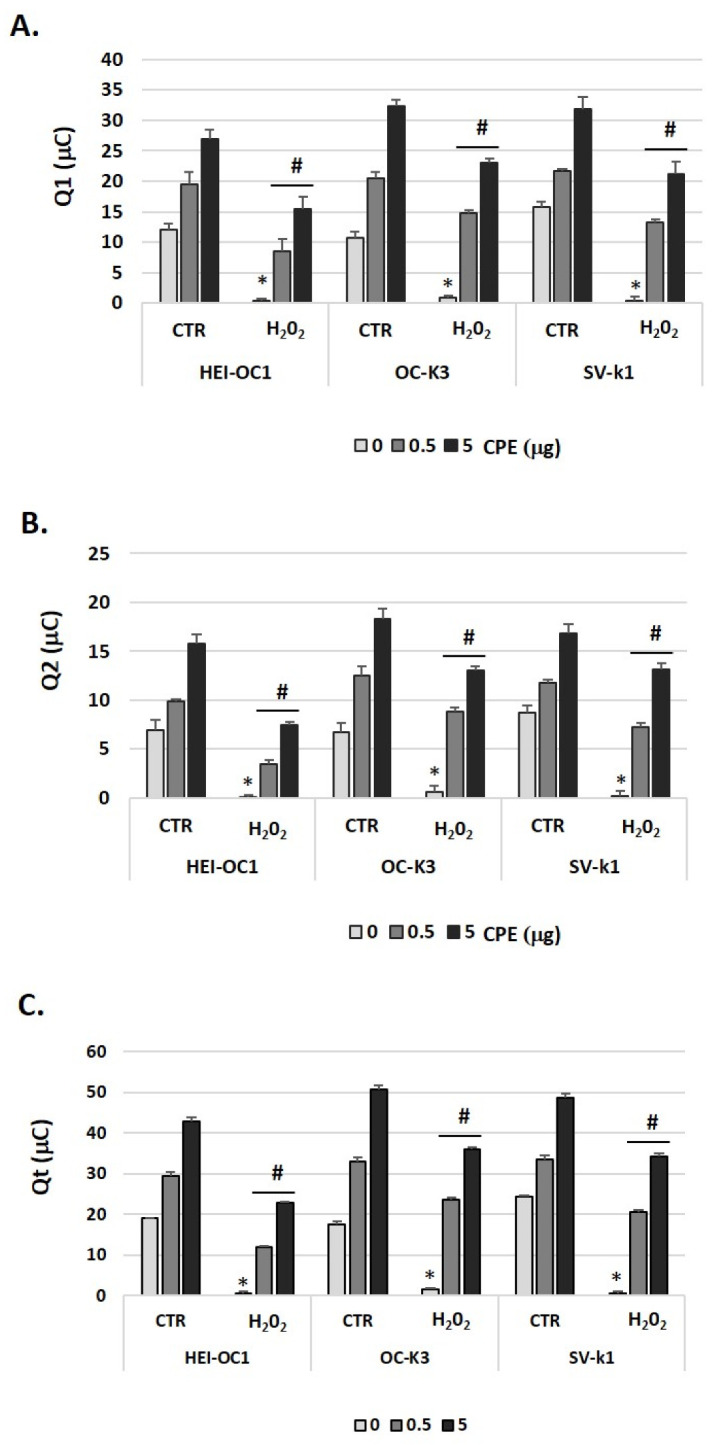
**Cocoa treatment modulated total antioxidant response in senescence auditory cells.** (**A**) Q1 (fast antioxidants), (**B**) Q2 (slow antioxidants), and (**C**) Qt (total antioxidant response: Q1 + Q2) (μC) were measured by e-BQC lab system in cell supernatants of HEI-OC1, OC-k3, and SV-k1 cells post-treatments. Experimental groups: CTR group (0, 0.5 and 5 μg/mL of CPE for 20 h) and H_2_O_2_ group (induced cellular senescence with 100 µM for 1 h in all groups: 0, 0.5 and 5 μg/mL of CPE for 20 h). The values represent the mean ± SD of micro-Coulomb (μC) in triplicate. * *p* < 0.05 vs. CTR group; # *p* < 0.05 vs. H_2_O_2_ group (*n* = 4).

**Figure 8 antioxidants-11-01450-f008:**
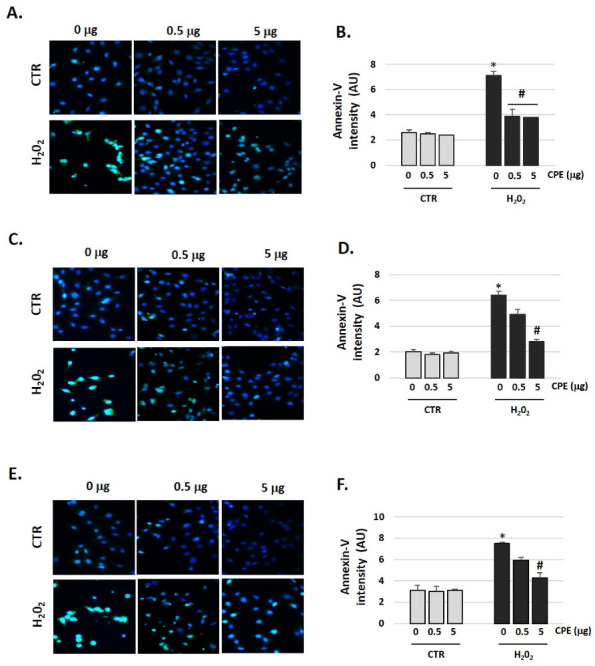
**Cocoa decreased production of Annexin-V in senescence auditory cells.** Representative fluorescence micrographs of immunostaining of Annexin-V in (**A**) HEI-OC1, (**C**) OC-k3, and (**E**) SV-k1 cell cultures post-treatments (×40). Nuclei were identified by DAPI (blue fluorescence), and Annexin-V immunoreactivity (green fluorescence). Representative images were taken at random in a blinded fashion. Bar graphs show the immunofluorescence intensity changes by Image J program for (**B**) HEI-OC1, (**D**) OC-k3, and (**F**) SV-k1 cells. Experimental groups: CTR group (0, 0.5 and 5 μg/mL of CPE for 20 h) and H_2_O_2_ group (induced cellular senescence with 100 µM for 1 h in all groups: 0, 0.5 and 5 μg/mL of CPE for 20 h). The values represent the mean ± SD Arbitrary units (AU) in triplicate. * *p* < 0.05 vs. CTR group; # *p* < 0.05 vs. H_2_O_2_ group (*n* = 4).

**Figure 9 antioxidants-11-01450-f009:**
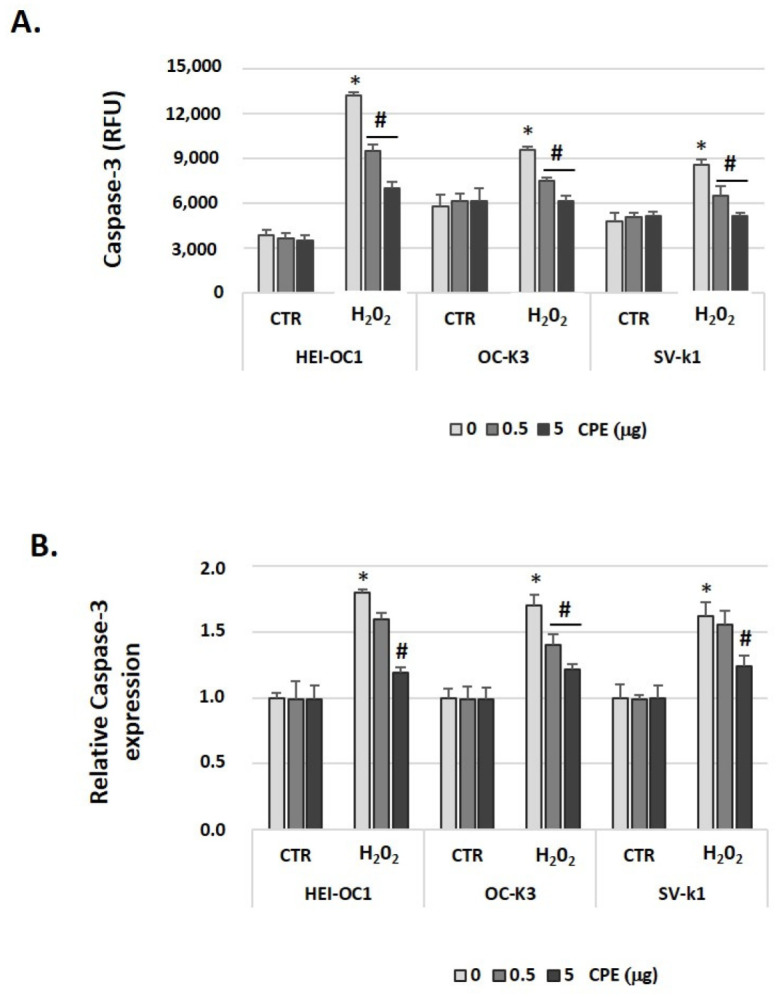
**Cocoa reduced the gene expression of caspase-3 in senescence auditory cells**. (**A**) Caspase-3 activity was measured by a fluorescent assay in HEI-OC1, OC-k3, and SV-k1 cell cultures post-treatments. (**B**) Gene expression of Caspase-3 in auditory cell cultures post-treatments. Experimental groups: CTR group (0, 0.5 and 5 μg/mL of CPE for 20 h) and H_2_O_2_ group (induced cellular senescence with 100 µM for 1 h in all groups: 0, 0.5 and 5 μg/mL of CPE for 20 h). The diagrams include the mean ± SD Fold change and Relative Fluorescence Units (RFU) in triplicate. * *p* < 0.05 vs. CTR group; # *p* < 0.05 vs. H_2_O_2_ group (*n* = 4). Reference genes: β-actin and GAPDH.

**Figure 10 antioxidants-11-01450-f010:**
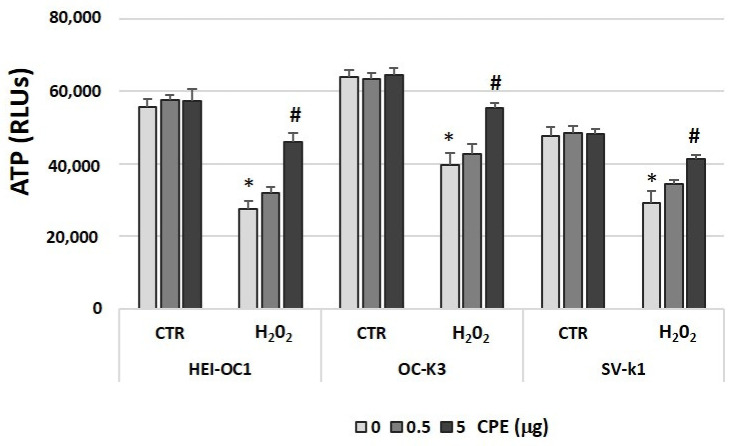
**Cocoa increased ATP levels in senescence auditory cells.** ATP levels were measured by a luminescence assay in cell lysates of HEI-OC1, OC-k3, and SV-k1 cells post-treatments. Experimental groups: CTR group (0; 0.5 and 5 μg/mL of CPE) and H_2_O_2_ group (induced cellular senescence with 100 µM for 1 h in all groups: 0; 0.5 and 5 μg/mL of CPE). The values represent the mean ± SD Relative Luminescence Units (RLU) in triplicate. * *p* < 0.05 vs. CTR group; # *p* < 0.05 vs. H_2_O_2_ group (*n* = 4).

**Figure 11 antioxidants-11-01450-f011:**
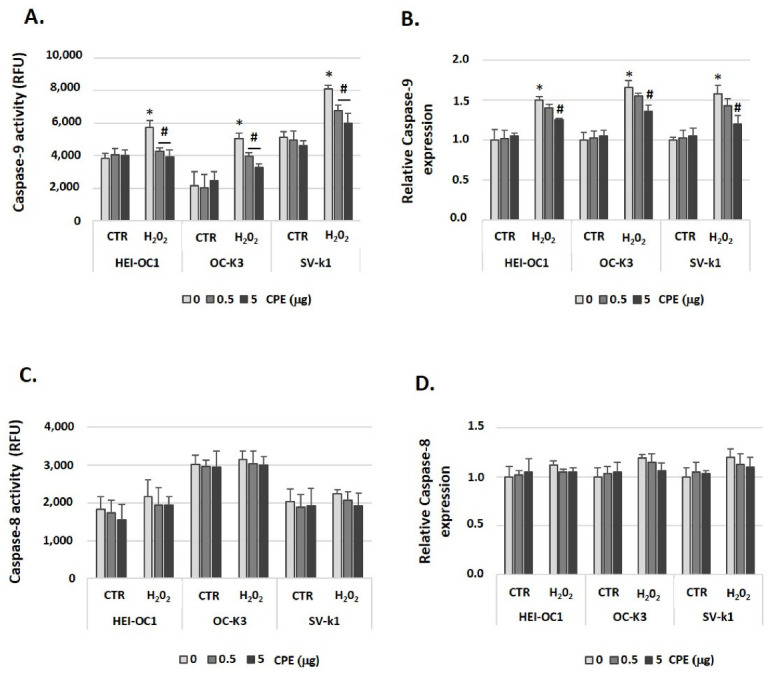
**Cocoa treatment impaired mitochondrial cell death pathway in senescence auditory cells.** (**A**) Caspase-9 and (**C**) Caspase-8 activity were measured by a fluorescent assay in HEI-OC1, OC-k3, and SV-k1 cell cultures post-treatments. Gene expression of Caspase-9 (**B**) and Caspase-8 (**D**) in auditory cell cultures post-treatments. Experimental groups: CTR group (0; 0.5 and 5 μg/mL of CPE) and H_2_O_2_ group (induced cellular senescence with 100 µM for 1 h in all groups: 0; 0.5 and 5 μg/mL of CPE). The diagrams include the mean ± SD Fold change and Relative Fluorescence Units (RFU) in triplicate. * *p* < 0.05 vs. CTR group; # *p* < 0.05 vs. H_2_O_2_ group (*n* = 4). Reference genes: β-actin and GAPDH.

**Figure 12 antioxidants-11-01450-f012:**
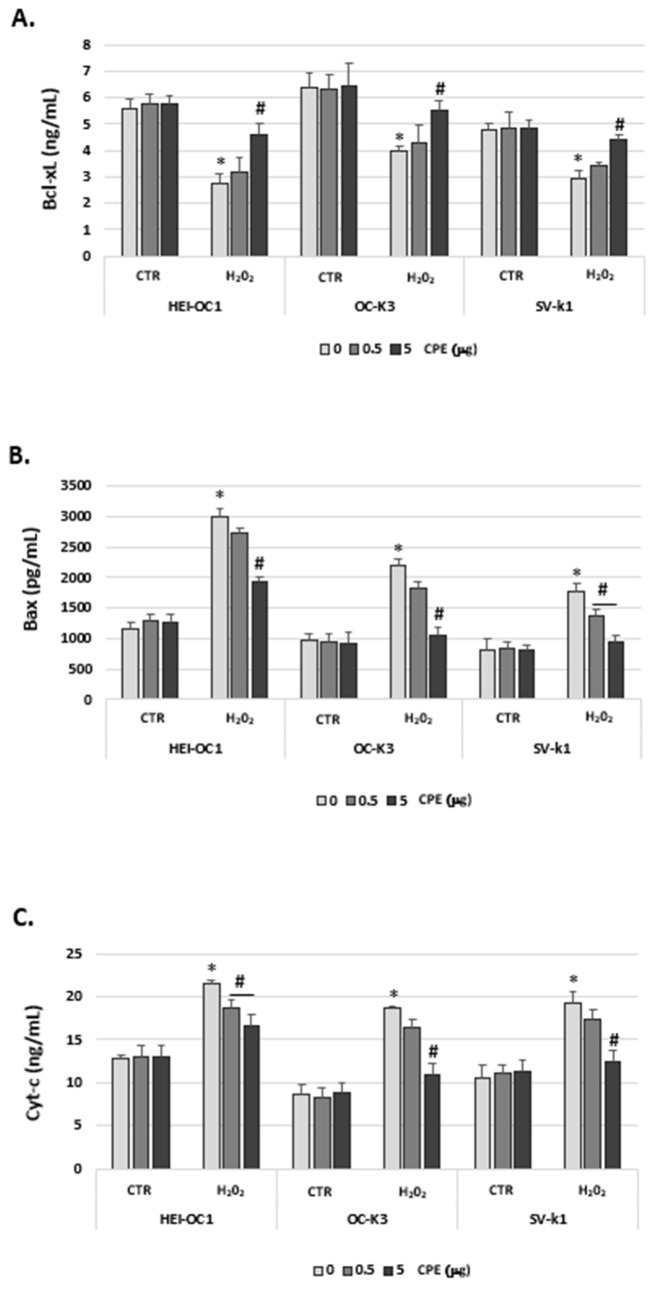
**Cocoa treatment impaired the mitochondrial-initiated cell death pathway in auditory senescence cells.** The Bcl-xL (**A**), Bax (**B**) and cytochrome c (**C**) levels were measured by a colorimetric assay in cell lysates of HEI-OC1, OC-k3, and SV-k1 cells post-treatments. Experimental groups: CTR group (0; 0.5 and 5 μg/mL of CPE) and H_2_O_2_ group (induced cellular senescence with 100 µM for 1 h in all groups: 0; 0.5 and 5 μg/mL of CPE). The values represent the mean ± SD ng/mL or pg/mL in triplicate. * *p* < 0.05 vs. CTR group; # *p* < 0.05 vs. H_2_O_2_ group (*n* = 4).

**Table 1 antioxidants-11-01450-t001:** Total polyphenol content in supernatants of cultured cells.

	*0* μg/mL	*0.5* μg/mL	*5* μg/mL	*10* μg/mL	*20* μg/mL
*CTR*	0	0.562	5.521	9.421	19.511
*H_2_O_2_ (100* μM*)*	0	0.423	4.524	8.289	18.116

## Data Availability

Data is contained within the article.

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
