# Peer review of "Preventive Effect of Cocoa Flavonoids via Suppression of Oxidative Stress-Induced Apoptosis in Auditory Senescent Cells"

_antioxidants, 2022, doi:10.3390/antiox11081450_

Round 1
Reviewer 1 Report
This is an important paper that proposes cocoa flavonoids as promising agents against senescence, via oxidattive stress in ARHL
The study is extremely interesting and thorough, but there are some points that require attention. 
Methods: the toxicity of cocoa polyphenol extract on cellular lines is not indicated, even if in Fig.2 is well shown that the last two concentrations induce loss of viability in all three cellular models. Please specify and clarify why the authors chose the lowest concentrations of CPE for the other experiments
Results: Table 1. It is very unusual to measure the polyphenol content in supernatants and not in the cells.
- The uptake of the polyphenols can be revealed only with a direct measure of CPE content in the inner of the cell. These data must be added to the paper.
- One of the main known mechanisms used by polyphenols to protect the cell from oxidative stress is the activation of the endogenous antioxidant defence systems (CAT, SOD, GPx, etc) and this action is mediated by the transcription factor NRF2. Adding this information to the paper would be desirable
Discussion: Despite the fact that the work has been carried out on three different cell lines and is rather accurate, it is necessary to add a sentence to the discussion on the limitation of the use of cell models in vitro.
Reviewer 2 Report
In this manuscript these authors have extended investigations on an in vitro cell senescence model of auditory cells using H2O2 to mimic the pro-oxidant effects of aging in the auditory system that they have published previously. H2O2 has been a commonly used inducer for stress-induced premature senescence in vitro which seems to replicate features of senescence in vivo. It appears to be a useful method to study aging in auditory cell lines.
In this study the authors demonstrate the efficacy of an extract of cocoa polyphenols against the oxidative stress and cellular damage induced by H2O2. The experiments appear to be properly designed and the interpretation of the data appear to be appropriate.
However, there are a few questions that should be considered by the authors.
Are the concentrations of polyphenols used in their experiments achievable by oral intake of cocoa by aging humans?
Why did the authors not examine the role of autophagy or mitophagy in their auditory senescence models as reported by Luo et al (Exp Gerontol 2912, 46, 860-867) and by Tsuchihashi et al (Oncotarget, 2015, 6, 3644-3655) for autophagy and Lin et al (Front Cell Neurosci 2019, 13, 550)?
Round 2
Reviewer 1 Report
The manuscript can be accepted in the present form
Author Response
Thank you